# Long Term Neurodevelopmental Outcomes after Sevoflurane Neonatal Exposure of Extremely Preterm Children: A Cross-Sectional Observationnal Study

**DOI:** 10.3390/children9040548

**Published:** 2022-04-12

**Authors:** Véronique Brévaut-Malaty, Noémie Resseguier, Aurélie Garbi, Barthélémy Tosello, Laurent Thomachot, Renaud Vialet, Catherine Gire

**Affiliations:** 1Department of Neonatology, North Hospital, University Hospital of Marseille, Chemin des Bourrelys, CEDEX 20, 13915 Marseille, France; veronique.brevaut@ap-hm.fr (V.B.-M.); aurelie.garbi@ap-hm.fr (A.G.); laurent.thomachot@ap-hm.fr (L.T.); renaud.vialet@ap-hm.fr (R.V.); catherine.gire@ap-hm.fr (C.G.); 2CEReSS—Health Service Research and Quality of Life Center, Faculty of Medicine, Aix-Marseille University, 27 Boulevard Jean Moulin, 13005 Marseille, France; noemie.resseguier@ap-hm.fr; 3CNRS, EFS, ADES, Aix Marseille University, 13915 Marseille, France

**Keywords:** extreme preterm, analgesia, Sevoflurane, long-term outcome, neurocognitive assessment

## Abstract

Sevoflurane, a volatile anesthetic, is used when extremely preterm neonates (EPT) undergo painful procedures. Currently, no existing studies analyze sevoflurane’s long-term effects during the EPT’s immediate neonatal period. Our primary objective was to compare the EPT’s neurocognitive development regardless of any sevoflurane exposure prior to 45 weeks corrected gestational age (GA). We analyzed those live discharges, less than 28 weeks GA, who were either exposed, unexposed, and/or multiply exposed to sevoflurane before 45 weeks GA. All data were obtained from a cross-sectional multicenter study (GPQoL study, NCT01675726). Children, both exposed and non-exposed to sevoflurane, were sampled using a propensity-guided approach. Neurological examinations (Touwen), cognitive and executive functions (WISC IV, NEPSY, Rey figure), and assessments when the children were between 7 and 10 years old, were correlated to their neonatal sevoflurane exposure. There were 139 children in the study. The mean gestational age was 26.2 weeks (±0.8) GA and the mean birth weight was 898 g (±173). The mean age of their evaluation was 8.47 years old (±0.70). Exposure to sevoflurane to the mean corrected age 27.10 (3.37) weeks GA had a significant correlation with cerebral palsy (adjusted odds ratio (aOR): 6.70 (CI 95%: 1.84–32.11)) and other major disorders (cerebral palsy and/or severe cognitive retardation) (aOR: 2.81 [95% CI: 1.13–7.35]). Our results demonstrate the possibility of long-term effects on EPT infants who had a sevoflurane exposure before 45 weeks corrected GA. However, these results will require further confirmation by randomized controlled trials.

## 1. Introduction

Improved neonatal intensive care over the last 30 years has increased survival rates of extremely preterm (EPT) infants [1]. Their care requires increasingly numerous and invasive procedures accompanied by extended intensive care stays. While it is unanimously accepted that pain management must be integral in the care of these newborns, up to 60–115 procedures per child can occur during their intensive care stay [2,3]. These painful and stressful procedures occur at a time when the infant’s central nervous system is maturing and can result in both short- and long-term consequences [4,5,6]. Practices are variable since no national or international consensus exists regarding analgesic management (indications, molecules, dosages for the type of invasive procedure, etc.) [3,7]. The lack of generalization and homogenization of practices is, in part, due to the anxiety generated by the literature’s abundant animal models on analgesics’ neurotoxic effects on the developing brain [8,9,10,11]. Among these treatments, volatile anesthetics, such as sevoflurane, are widely used in pediatrics because of their pharmacokinetic properties. This drug provides rapid anesthesia without needing a venous approach and enables a quick, post-procedural wake-up [12,13].

Three recent studies, including a randomized controlled trial, reported reassuring findings on the long-term effects of its use in infants: a Randomized Control Trial (RCT) General Sevoflurane Anesthesia (GAS) study compared to spinal anesthesia study; MASK (Mayo Anesthesia Safety in Kids) study when associated with other anesthetics; and PANDA (Pediatric Anesthesia & Neurodevelopment Assessment) study [14,15,16].

The GAS study included preterm infants over 26 weeks GA but for whom anesthesia was performed around 45 (±4.5) weeks corrected GA. Results in a predominantly male study population indicated that slightly less than 1 hour of general anesthesia in early infancy does not alter neurodevelopmental outcomes at age 5 years as compared to awake–regional anesthesia [14].

The MASK study concerned only term newborns. Anesthesia exposure was not associated with deficits in general intelligence but multiple exposures were associated with a pattern of changes in specific neuropsychological domains associated with behavioral and learning difficulties [15].

The PANDA study concerned twins born after 36 weeks GA, both exposed and unexposed to anesthesia and performed after the first month of life [16]. There were no statistically significant differences in the full-scale intelligence quotient (FSIQ) later in childhood.

To our knowledge, no clinical study has analyzed Sevoflurane’s long-term effects on EPT populations when used during their immediate neonatal period before 45 weeks corrected GA.

We hypothesized that sevoflurane’s use during this population’s critically vulnerable synaptogenesis period affected their normal brain development, and our objective was to compare the neurocognitive development of prematurely born children, according to exposure to sevoflurane, before 45 weeks corrected GA.

## 2. Method

### 2.1. Subjects and Methods

#### 2.1.1. General Framework

This was a cross-sectional, multicenter study of school-aged EPT children who were hospitalized in two French level III facilities (Marseille, France). Both hospitals are authorized to care for EPT infants less than 28 weeks GA. These children, alive at discharge and with complete data concerning their exposure to anesthesia, were evaluated when they were between 7 and 10 years of age. Each received a clinical examination and an assessment of their motor and cognitive functions during a GPQoL study’s time. Motor skills were assessed by the Touwen Infant neurological examination [17]. A psychometric assessment was performed, using the WISC-IV the Rey’s figure: a short perceptual organization and memory test, and subtests of NEPSY (NEuroPSYchological assessment) evaluating attention and executive functions [18,19,20].

#### 2.1.2. Participants

Our infant population, obtained from the two participating neonatal centers, were all eligible for the GPQOL study as published in 2018 [21].

Inclusion criteria were EPTs born prior to 28 weeks GA, between 1 January 2004 and 31 December 2006, and hospitalized in one of the two level III participating facilities. Infants all had received anesthesia during their hospitalization and were alive at discharge. The exclusion criteria included medical records that had no neonatal anesthesia data collection and absence of surgery to avoid indication bias. We chose to exclude the patients without any anesthesia and did not use them as a control group since these children were not comparable to those who required anesthesia. We compared the perinatal characteristics of these children without any anesthesia and even if the gestational age and birth weights were comparable to those exposed, they presented a less important severity: less nosocomial infections (46 vs. 60%; *p* = 0.014), less red cells transfusion (46 vs. 85%; *p* = 0.000), and shorter mean length of stay in hospital in days (85 vs. 100%; *p* = 0.010). Our hypothesis is that the indication for anesthesia reflects the severity of the children’s condition. It therefore seemed more appropriate to exclude them from this analysis to avoid indication bias and increase the quality of our work.

### 2.2. Ethics

These newborns were included in a multi-center study (GPQoL study) to evaluate the quality of life of school-aged children, and to assess their cognitive functions and neurological examinations when they were between 7 and 10 years old. Parental informed consents were obtained for each participant. This national study was approved by the French Patient Protection Committee (Number IDRCB 2012-A00193-40, 25 September 2013) and was submitted to the National Committee for Information Technology and Freedom (CNIL *Commission nationale de l’informatique et des libertés*) under number 1427029. The ClinicalTrials.gov identifier is NCT01675726.

### 2.3. Data Collection

#### 2.3.1. Neonatal and Anesthesia Data

Clinical data on Sevoflurane neonatal anesthesia were extracted from the medical records of each child by a single investigator (VBM). The investigator did not know the child’s state nor its neurocognitive assessment.

Data collection included the exposure duration to sevoflurane, the child’s corrected and postnatal age, when the anesthetic agent was administered, and the indications for using sevoflurane.

The medical record analyses did not provide any reliable recovery of inhaled concentrations of administered anesthetic. However, the anesthetic objective was always the same (absence of pain, closed eyes, loss of motor response to gentle handling, and muscle tone) and anesthesia was administered according to the practices described in a previously published article [22]. The administered anesthetic’s molecule, calibrated dose per kilograms of weight, and the duration of exposure were also noted.

#### 2.3.2. Perinatal Data

The child’s demographic characteristics were retrieved by another investigator (CG) who is associated with the princeps study (GPQoL). These characteristics included gestational age (GA) at birth, gender, birth weight (BW), parental socio-economic level (SEL), neonatal complications secondary to prematurity: bronchopulmonary dysplasia (BPD), necrotizing enterocolitis (NEC), patent ductus arteriosus (PDA), anemia, abnormal cranial ultrasound or cerebral magnetic resonance imaging (MRI) between 36–40 weeks GA-corrected age, and retinopathy of prematurity (ROP).

The parental Socio Economic Level (SEL) was assessed using the Family Affluence Scale (FAS) and BPD was defined as the need for ventilator support at 36 weeks GA-corrected age [23]. Newborns with clinical signs of enteritis and/or pneumatosis viewed with standard radiography and/or abdominal ultrasound were classified as having NEC. Children requiring medical (anti-prostaglandin) or surgical closure of the ductus arteriosus were defined as having PDA. Anemia was defined as the requirement for at least one transfusion of red blood cells during the neonatal period. The cranial ultrasound was considered abnormal if there was a high-grade intraventricular hemorrhage (≥3). The cerebral MRI was considered abnormal if there was an abnormality of the basal ganglia or diffuse abnormality of the white or gray matter. Surgery was defined as any procedure requiring general anesthesia in the operating room. The surgical indications (NEC, PDA, ROP, and inguinal hernia) were obtained from the medical records and the duration of the procedure was noted. The others indications for anesthesia, apart from surgery, were central line placement (piccline) or nasotracheal intubation and were also noted.

#### 2.3.3. Evaluation between Seven and Nine Years Old

Prior to the completion of the neurocognitive evaluation, each subject received a complete clinical evaluation, which included chronic conditions, usual treatments, their academic levels along with the need of possible assistance, paramedical care (physiotherapist, speech therapy), measurements, and neurological examinations involving gross and fine motor skills by Touwen Infant neurological examination [17]. Based on these clinical evaluations, cerebral palsy was defined according to Bax and severity was defined according to the Gross Motor Function Classification System (GMCFS) and bimanual motor function (BMF) classification [24,25,26].

Cerebral palsy was defined as any abnormality of tone or posture. This broad definition therefore includes cerebral palsy’s traditional definition as well as children with only very moderate cerebral palsy (Type 1 and 2 GMCFS and/or BMF) who have thus benefited from neuro-psychological assessments and neuro-motor examination. Disability was defined according to the mean of the FSIQ and the results of the Touwen Infant neurological examination. (i) No disability: an FSIQ ≥ 89 and a normal Touwen; (ii) Mild disability: an FSIQ <89 and ≥79 and/or an abnormal Touwen; (iii) Moderate disability: an FSIQ <79 and ≥65 independent of the Touwen results; (iv) Severe disability or mental delay: an FSIQ < 65 independent of the Touwen results or autism, according to the DSM IV classification or a severe CP not eligible for neurocognitive assessment [27].

A “specific neurocognitive impairment” was considered if at least one of the following five specific neuropsychological mental illness disorders (DSM IV classification of mental diseases) was noted [28]. (i) Language delay if verbal comprehension index (VCI) was <85; (ii) Motor ideomotor dyspraxia if the Touwen test found a complex coordination disorder and a perceptual reasoning index (PRI) as <85 (thus a significant proportion of children with predominantly ideomotor dyspraxia were also classified as GMFCS type I or 2 and/or in BMF type I or 2); (iii) Visuo-spatial integration delay if PRI was <85 and there was a poor copy of Rey’s figure; (iv) Dysexecutive disorders if the working memory index (WMI) was <85 and/or cognitive inhibitions <10th percentile and/or fluidity of patterns was <8 and/or the tower was <8; (IIIII) Attention deficit if auditory and/or visual selective attention was <8 and a Processing Speed Index (PSI) was <85 [25,26].

#### 2.3.4. Outcomes

The primary outcomes were binary outcomes defined by surviving with a total FSIQ lesser than −1 standard deviation (SD) (i.e., ≤85) at school age.

Secondary outcome was constituted by a long-term relationship of sevoflurane exposition with neurocognitive impairment or cerebral palsy.

### 2.4. Statistical Analysis

We analyzed the relationship between outcome using the propensity score approach to control for observed confounding factors that might influence both group assignments (exposed and non-exposed to sevoflurane) [29]. Propensity score was defined as the infants’ probability of being exposed to sevoflurane based on their individual covariates. Because on the small number of children who were exposed to anesthesia without sevoflurane, we chose to match two cases per control to increase the number of subjects in our analysis and increase its power. In our study, the propensity score was estimated by a multivariate logistic regression model. Propensity score matching 2:1 was adopted with pre-selected explanatory variables: antenatal steroids, GA, gender, chronic lung disease, surgery requirement, nosocomial infection, PDA, anemia, and retinopathy of prematurity (Appendix A).

The association between sevoflurane and/or other anesthetic and neurocognitive outcomes (cerebral palsy, neurocognitive impairment, language delay, attention deficit, visuo-spatial integration delay, ideomotor dyspraxia, and dysexecutive disorders) was first studied by univariate analysis. Logistic regression analysis was used to estimate adjusted odds ratios with their 95% confidence interval when considering the following qualitative outcomes: cerebral palsy, global cognitive deficit, major disabilities, and FSIQ ≤ 85. Firth’s correction was applied by performing Firth’s penalized-likelihood logistic regression to take into account the small number of events when necessary [29,30]. Linear regression models were used to estimate beta coefficients with their 95% confidence interval when considering the following quantitative outcomes: FSIQ, VCI, and WMI. No data selection procedure based on statistical criteria was performed. Sevoflurane exposure was systematically included in multivariate models. Multivariate models were proposed to assess the impact of exposure to sevoflurane on neurocognitive outcomes, using the same strategy as described above for the analysis of sevoflurane exposure with propensity score. All analyses were performed using R software version 3.0.3 (R Foundation for Statistical Computing, Vienna, Austria. http://www.R-project.org/ (accessed on 2 November 2021)). The R package logistf was used for Firth’s penalized-likelihood logistic regression (R package version 1.21. https://cran.r-project.org/ (accessed on 2 November 2021)). All tests were performed two-sided, and for all analyses a *p*-value < 0.05 was considered statistically significant.

## 3. Results

### 3.1. Population

A total of 103 children were included in the study with 85 (82.5%) being exposed to at least one anesthesia with sevoflurane before 45weeks GA and 18 (17.4%) with anesthesia but without sevoflurane during this period (Figure 1).

The population’s clinical long-term outcomes are shown in Table 1. There were 84 children who received a neurocognitive assessment at an average age of 8.68 (±0.49) years. The mean FSIQ was 84.80 (±18.41), 10 (9.3%) children had severe cerebral palsy, and 37 (36.3%) a severe disability i.e., severe cerebral palsy and/or mental delay (FSIQ < 70) and/or autism.

The baseline characteristics according to exposure to sevoflurane are presented in Table 2. Birth weight was significantly higher in the exposed group (861.64 vs. 739.72 g; *p* = 0.020). The exposed group had significantly less retinopathy of prematurity (11.76 vs. 33.33%; *p* = 0.033) and more surgery requirement (42.35 vs. 11.11%; *p* = 0.013). The time of exposition was 27.10 (3.37) weeks corrected GA. The indications for anesthesia were surgery, central line placement (piccline), or nasotracheal intubation. The list and the frequency of each indication in each group is shown below and included in Table 3

### 3.2. Propensity Score (PS) Matched Analysis

After PS matching, less significant differences were observed among these variables, above all, any indication bias (more surgery in the group exposed in the overall cohort). Birth weight stayed significantly higher in the exposed group (Table 2).

The neonatal characteristics of the 18 unexposed children in the neonatal period and the 36 exposed children became comparable in terms of their GA, exposures to antenatal corticosteroids, adaptations to extra uterine life, and their socio-economic conditions.

### 3.3. Comparison Exposed/Unexposed in Matched Group

The number of children with FSIQ ≤ 85 did not differ between the exposed and unexposed groups (8 (50%) vs. 20 (71%); OR 2.41 [0.70–8.61]; *p* = 0.16) (Table 4). Mean FSIQ was also not statistically different between the two groups but there was a trend toward a lower FSIQ in the exposed group ((75.93 (±15.86) vs. 86.06 (±22.51); *p* = 0.05) in the PS model (Table 5).

In the matched group, there is a significant difference between exposed and unexposed to sevoflurane only for modality anesthetic and analgesic treatments. The mean GA at first use of sevoflurane anesthesia was 27.10 (3.37) weeks GA corrected and 6.89 (23.7) days, respectively (Table 3 and Table 5). There were no differences between the groups in the number of abnormal cranial US or MRIs, or in length of hospitalization.

The long-term outcome of exposed sevoflurane children is significantly correlated with more severe disorders: cerebral palsy and/or mental retardation. The correlation existed also significantly for impairments, gesture planning, and/or visual perception.

The PS was estimated by a multivariate logistic regression model; 2:1 propensity score matching was adopted by matching: nosocomial infection, anemia, surgery, bronchoplumanary dysplasia, patent ductus arteriosus, and retinopathy of prematurity. After matching, a logistic regression model was performed to compare the outcome measures between sevoflurane and non-sevoflurane groups. Firth’s correction was applied by performing Firth’s penalized-likelihood logistic regression to take into account the small numbers.

### 3.4. Effect of Exposure to Sevoflurane in the Neonatal Period for the Long Term Outcome in Multivariate Models and with the Propensity Score

The neurocognitive outcomes according to sevoflurane exposure in univariate and multivariate models and with the propensity score are detailed in Table 4.

Exposure to sevoflurane had a significant link to cerebral palsy (aOR: 5.09 [CI 95%: 1.18–32.99]), severe disability (3.75 [CI 95%: 1.10–15.26]), visuo-spatial integration delay (6.97 [CI 95%: 1.21–90.67]), and ideomotor dyspraxia (6.97 [CI 95%: 1.21–90.67]) with multivariate models.

With the propensity score, the exposure to sevoflurane was significantly associated with the same sequelae and with a processing speed index ≤ 85 (aOR: 4.20 [CI 95%: 1.24–15.41].

### 3.5. Effect of Exposure to Sevoflurane in the Neonatal Period Alone or in Association for the Long-Term Outcome in Multivariate Models and with the Propensity Score

Comparison of children who received sevoflurane alone (*n* = 43) compared with those who received sevoflurane in combination with other anesthetics (*n* = 42) suggests that the combination of anesthetics potentiates the long-term effects of sevoflurane (Table 6).

## 4. Discussion

Although our results do not demonstrate that exposure to sevoflurane, used in EPT neonates (before 45 weeks GA-corrected age), resulted in a significant decrease in FSIQ, they do show an association between sevoflurane exposure with an increased occurrence of severe neurologic disorders such as cerebral palsy, or severe disability or moderate impairment such as visuo-perception delay and/or ideomotor dyspraxia.

Sevoflurane is widely used in neonates born premature for invasive procedures (intubation, central catheterization), cerebral MRI or fundus examinations, or for surgery [22,31,32,33,34].

The long-term effects of anesthetics on neonates and infants vary, based on the anesthetic used, the number and duration of exposures, the child’s age, the age at the time of exposure, and the type of assessment (the age and instruments used) [14,16]. Recently, a randomized controlled trial reported reassuring data on the use of sevoflurane during a single exposure [14]. However, our study does not reflect actual exposures since children were EPT when subjected to many invasive procedures requiring a repeated use of anesthetics.

As is widely demonstrated in animals, the number and duration of exposures to the anesthetic impacts the neurotoxic effects [35,36,37,38]. The reassuring results of the GAS, MASK, and PANDA studies concern only a single exposure to sevoflurane [14,15,16]. The mean exposure times were all under 2 hours: 54 min, 67 min, and 84 min, respectively [14,15,16]. The mean exposure times in these studies were 54 min for GAS, 67 for MASK, and 84 for PANDA, all less than 2 h [14,15,16].

Our study’s mean total exposure time is 172 min, which reflects the reality of the exposure in these EPT newborns. However, in the MASK study, a long exposure time ( > 120 min) was also significantly associated with the development of learning and ADHD disorders [15]. Recent animal studies found similar results [36] and are in agreement with our own findings. In the PANDA study, this association was not found [16]. However, the PANDA study only involved a single exposure and the children having anesthesia for more than 120 min were few (*n* = 17). The MASK study demonstrates that exposure to several anesthesia sessions before 3 weeks old significantly impacts learning disabilities (reading, writing and mathematics) as well as attention deficit hyperactivity disorder (ADHD) [15]. This finding has also been demonstrated in animal studies [36].

The gestational age at birth and the child’s age when exposed to the anesthetic also play a crucial role. Unlike our study, there are no published studies on the effects of Sevoflurane exposure for EPT infants during their immediate neonatal period.

The overall FSIQ figures in our study for the exposed and non-exposed groups are much lower than those in the PANDA study. 16 That can be explained by our EPT children population, in which the average FSIQs are known to be lowered by 1.5 points for each week below 33 weeks GA [39]. Our results are in agreement with the results of a cohort of preterm infants evaluated under the same conditions at 6.5 years [40]. Conversely, the PANDA figures do not appear to be representative of the general population and raises questions about the generalization of their results [16].

The simultaneous use of several anesthetics (28.06% of newborns in our cohort) can limit interpretation of our results. Our results indicate that the use of sevoflurane alone is less harmful than its use in combination with other anesthetics, but is significantly associated with ideomotor dyspraxia. In the GAS study, sevoflurane was exclusively used [14]. The potentiation of the neurotoxic effects of multiple anesthetics simultaneously used has been demonstrated in animals [10,38]. However, the concurrent use of several anesthetics occurred in the PANDA studies (27% of children received gas anesthesia and 71% opioid as a complement) [16]. In the MASK study, 10% of children received gas anesthesia and 44% opioid supplementation [15]. Our data, and those of the last two studies, reflect the reality of the exposure of these patients for whom sevoflurane anesthesia alone did not allow sufficient sedation and required additional treatment, in particular opiates. However, the multivariate and propensity score analysis results confirm that the use of another anesthetic is not a confounding factor. Finally, the possibility that some children in our cohort may have been exposed to sevoflurane or other anesthesia between the initial neonatal period (after 45 Weeks GA) and the assessment period was not accounted for in our analysis. However, reassuring data from recent children’s gaseous anesthesia studies indicate that our findings would not have been affected [14,15,16].

Understanding the mechanisms of anesthesia-related damage in the systems that mediate sensory input versus learning-related networks is an important aspect to building a more complete understanding of the effects of anesthesia in neonates and above all in preterm. Anesthesia can induce structural, functional, and compensatory changes in both systems, which can be influenced by a wide variety of factors. Changes in myelination induced by anesthesia exposure appear less significant compared to the neurodegeneration observed in the gray matter and in a variety of brain regions. In our study, this neurodegeneration is more important when two anesthetics are used but still exists when sevoflurane is used alone.

Disproportionate cell death between excitatory and inhibitory cells induced by anesthesia exposure can lead to a long-term shift in the excitatory/inhibitory balance, which affects both learning-specific networks and sensory systems. Furthermore, anesthesia may directly affect not only synaptic plasticity, which is especially critical to learning acquisition, but also sensory adaptation to stimulation. Sensory systems appear to have better ability to compensate for damage than learning-specificity networks. However, as it has been shown that severity of the brain damage depends on the anesthesia protocol, it is possible that sensory dysfunction could be produced by a combination of different damage [39]. Notably, the postnatal development of the learning and sensory systems are different, and the timeline of development is not necessarily equivalent in humans and laboratory animals. Future studies of early anesthesia exposure in humans especially in VPT should account for these developmental differences conclusions about the impact on sensory systems.

Our study’s limitations reside firstly in its retrospective collection of anesthesia data, since the percentages of sevoflurane during each anesthesia was not accessible. Animal data support a dose-dependent effect of anesthetics on the developing brain [37,41]. However each child was on a continuously monitored ventilator in order to ensure the stability of their vital signs during each sevoflurane anesthesia. There was little variation in the percentage of sevoflurane used, as reported in our team’s previous study for the installation of a central catheter on 24 newborns with a median gestational age of 31 weeks GA [31]. By and large, it seems unlikely this limitation influenced our study’s results.

A second limitation is the high proportion of cerebral palsy found in our population of 103 children (26.73%), as well as that of 85 children exposed to sevoflurane in the neonatal period (30.12%). These figures are much higher than those found in populations of this term born at the same time when cerebral palsy rates vary from 7 to 10% depending on the studies. This high proportion of cerebral palsy may contrast at first reading with the cognitive function results of these children with a mean total FSIQ of −1 DS (85.3 and 84.5 in exposed and not exposed groups, respectively), which are comparable to those found in the literature (mean total IQ of 83.4 in the Swedish EXPRESS cohort, a study in which the cerebral palsy rate was only 9.5%) [40]. This is due to the choice of our very broad definition of cerebral palsy corresponding to any child with a tone or posture anomaly or an anomaly o planification of movement. A significant proportion of children with predominant ideomotor dyspraxia were therefore classified as GMFCS type I and/or BMF type I. This definition therefore includes cerebral palsy as usually described in the studies cited above, as well as children with only very moderate cerebral palsy (type 1) who have benefited from neuro-psychological assessments and neuro-motor examination. This explains why our fairly high figures for cerebral palsy are not comparable to those in the literature.

A third limitation is the variety of indications for anesthesia. Because varying pathologies can themselves influence outcomes, this can become a bias in our interpretation of results [42]. However, the multivariate analysis results allowed us to conclude that surgery, regardless of the indication, is neither a confounding factor nor does it explain the high proportion of cerebral palsy and major disorders among those children exposed to Sevoflurane.

The interpretation of these results must also take into account when the child was evaluated and the tools used for measuring cognitive outcomes. Using clinical diagnoses or academic outcomes in retrospective studies, without using standardized neuropsychological FSIQ type evaluations, limits some studies’ interpretation of results [15]. Our evaluation chose a standardized FSIQ range of 7–9 years in order to provide a reliable and accurate analysis of these children’s long-term outcomes.

Our results demonstrate an association between neonatal sevoflurane exposure on extremely premature infants born before 45 weeks GA-corrected age and long-term neurologic effects. Nonetheless, these findings need to be confirmed by controlled randomized studies.

These frail newborns have an increased survival rate and also require a fair amount of invasive care. It is urgent to conduct studies and find options, pharmacological or otherwise, to subdue their pain without affecting their long-term fate [1,43].

## Figures and Tables

**Figure 1 children-09-00548-f001:**
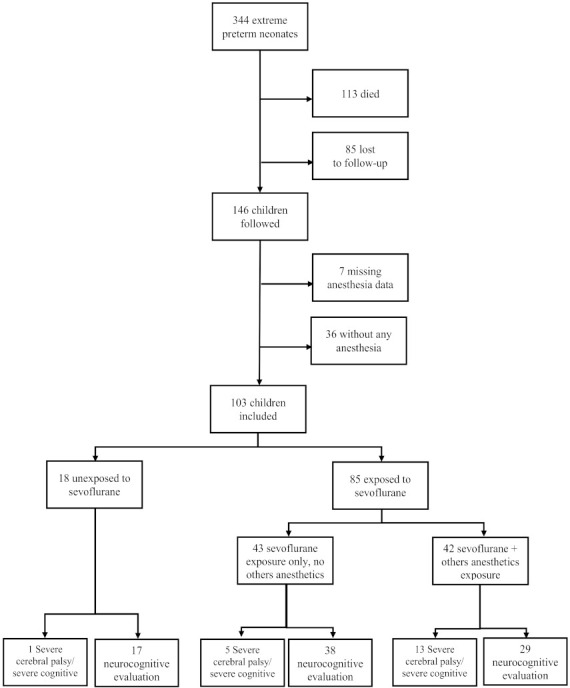
Flow-chart. GA: birth term (weeks gestational age).

**Table 1 children-09-00548-t001:** Population description and neurocognitive behavior outcome evaluation outcomes between 7 and 9 years old (*n* = 103).

	Number (*n*)	% or Mean (±SD)
* **Neurocognitive evaluation** *		
Mean age at testing in years (±SD)	91/103	8.68 (±0.49)
*WISC-IV*		
Mean FSIQ (±SD)	82/84	84.80 (±18.41)
FSIQ ≥ 115	5/82	6.10%
FSIQ ≤ 85	43/82	52.44%
Mean VCI (Verbal Comprehension Index) (±SD)	83/84	95.04 (±19.75)
Mean PRI (Perceptual Reasoning Index) (±SD)	84/84	86.18 (±15.99)
Mean WMI (Working Memory index) (±SD)	83/84	87.30 (±15.27)
WMI ≤ 85	32/83	38.55%
Mean PSI (Processing Speed Index) (±SD)	84/84	84.89 (±14.97)
PSI ≤ 85	45/84	53.57%
*NEPSY*		
Mean executive function/planning score (Tower) (±SD)	84/84	10.60 (±2.72)
Mean auditory attention score (±SD)	81/84	9.19 (±1.42)
Mean visual attention score (±SD)	83/84	9.02 (±2.95)
Inhibition score (Statue) ≤ 10th percentile	2/82	2.44%
Mean design fluency score	84/84	7.04 (±2.67)
* **Neurocognitive outcome** *		
^1^ Cerebral palsy (CP) (10 Severe and 17 Minor CPs)	27/101	26.73%
Severe CP: GMFCS and/or BMF 3 or 4	10/101	10%
Minor CP: GMFCS and/or BMF type < or = 2	17/101	16.8%
^2^ Severe disability	37/101	36.63%
^ **3** ^ ** Specific neurocognitive impairment**	59/84	70.24%
Dysexecutive disorders	28/83	33.73%
Attention deficit	20/82	24.39%
Visuo-spatial integration delay	22/84	26.19%
Motor ideomotor dyspraxia	25/83	30.12%
Language delay	42/88	47.73%

Data expressed as % or mean (±SD). SD: standard deviation; WISC-IV: Wechsler Intelligence Scale for Children—Fourth Edition; FSIQ: Full Scale Intellectual Quotient; VCI: Verbal Comprehension Index; PRI: Perceptual Reasoning Index; WMI: Working Memory index; PSI: Processing Speed Index. ^1^ Cerebral palsy was defined as any abnormality of tone or posture according to the definition of Bax [24]. This broad definition therefore includes cerebral palsy as usually defined as well as children with only very moderate cerebral palsy (type 1 and 2 Gross Motor Function Classification System (GMCFS) and/or bimanual motor function (BMF) who have thus benefited from neuro-psychological assessments and neuro-motor examination. ^2^ Disability was defined according to the mean of the Full-Scale Intelligence Quotient (FSIQ) and the results of the Touwen Infant neurological examination. (i) No disability: an FSIQ ≥ 89 and a Touwen normal; (ii) Mild disability: an FSIQ < 89 and ≥79 or a Touwen abnormal; (iii) Moderate disability: an FSIQ < 79 and ≥65 independent of the Touwen result; (iiii) Severe disability: mental delay and FSIQ < 65 independent of the Touwen result or autism, according to the DSM IV classification, or a severe CP not eligible for neurocognitive assessment. ^3^ A “specific cognitive impairment” was considered if at least one of the following five specific neuropsychological mental illness disorders (DSM IV classification of mental diseases) was noted: (i) Language delay if verbal comprehension index (VCI) <85; (ii) Motor ideomotor dyspraxia if the Touwen test found a complex coordination disorder and a perceptual reasoning index (PRI) was <85; (iii) Visuo-spatial integration delay if PRI was <85 and there was a poor copy of Rey’s figure; (iv) Dysexecutive disorders if the working memory index (WMI) was <85 and/or motor inhibitions <10th percentile and/or fluidity of patterns was <8 and/or the Tower was <85; (v) Attention deficit if auditory and/or visual selective attention was <8 and a Processing Speed Index was (PSI) <85.

**Table 2 children-09-00548-t002:** Baseline characteristics according to exposure to sevoflurane.

	Overall (*n* = 103)				PS ** Matched Cohort (*n* = 54)			
	Non-Exposed (*n* = 18) (%)	Exposed to Sevoflurane (*n* = 85) (%)	*** Standardized Difference	*p*-Value	Non-Exposed (*n* = 18)	Exposed to Sevoflurane (*n* = 36)	*** Standardized Difference	*p*-Value
* **Antenatal data** *								
Antenatal steroids	16 (88.89)	82 (96.47)	0.2	0.20	16 (88.89)	33 (91.67)	0.094	>0.99
* **Perinatal data** *								
Male gender	9 (50.00%)	45 (52.94)	**0.29**	0.82	9 (50.00%)	17 (47.22)	0.056	0.84
Mean GA at birth in Weeks GA (±SD)	25.89 (±0.96)	26.27 (±0.79)	**0.43**	0.10	25.89 (±0.96)	26.31 (±0.71)	**0.49**	0.11
Mean BW in grams (±SD)	739.72 (±143.78)	861.64 (±186.54)		**0.020 ***	739.72 (±143.78)	877.34 (±175.50)		**0.0094 ***
* **Neonatal morbidities** *								
CLD	11 (61.11)	43 (52.44)	0.17	0.50	11 (61.11)	20 (66.67)	**0.69**	0.697
Nosocomial infections	12 (66.67)	51 (60.00)	**0.98**	**0.19**	12 (66.67)	24 (66.67)	.0.00	>0.99
PDA	11 (61.11)	55 (64.71)	**0.77**	**0.074**	11(61.11)	25(69.44)	0.17	0.54
Retinopathy of prematurity (all stages)	6 (33.33)	10 (11.76)	**0.53**	**0.033 ***	6(33.33)	7(19.44)	**0.31**	0.31
Surgery required (all indications)	2 (11.11)	36 (42.35)	**0.75**	**0.013 ***	2(11.1)	5(13.89)	0.084	>0.99

Data expressed as *n* (%) or mean (±SD); SD: standard deviation; Abb: IUGR: Intrauterine growth retardation; FAS: Family Affluence Scale; GA: gestational age; Weeks GA: weeks of amenorrhea; BW: birth weight; SGA: Small for gestational age; CLD: chronic lung disease; NEC: necrotizing enterocolitis; PDA: Patent ductus arteriosus; MRI: Magnetic resonance imaging; only 88 cerebral MRI performed: 21 in non-exposed group and 67 in sevoflurane-exposed group; *: *p* < 0.05. ** The PS was estimated by a multivariate logistic regression model. A 2:1 propensity score matching was adopted by matching: nosocomial infection, anemia, surgery, severe bronchopulmonary dysplasia, patent ductus arteriosus, retinopathy of prematurity, term birth, gender. *** Standardized Difference: Standardized difference before/after matching (the acceptability threshold for standardized differences is generally set at 0.10 or 0.20).

**Table 3 children-09-00548-t003:** Comparison modality of sevoflurane, analgesia, and anesthesia treatment between groups (overall cohort and with the propensity score (PS)).

	Overall Cohort (*n* = 103)			PS-Matched Cohort (*n* = 54)		
	Non-Exposed (*n* = 18)	Exposed to Sevoflurane (*n* = 85)	*p*-Value	Non-Exposed (*n* = 18)	Exposed to Sevoflurane (*n* = 36)	*p*-Value
**Treatment with Sevoflurane**						
Corrected age (Week GA) exposure		28.61 (5.07)			27.28 (3.36)	
Age exposure (Days)		16.82 (37.29)			6.89 (23.7)	
Mean Exposure time (minutes)		151 (138)			149 (109)	
**Treatment with anesthesia * **						
Other anesthesia	18 (100)	42 (49.4)	**0.0001**	18 (100)	16 (44.4)	**0.0001**
Corrected Weeks GA exposure anesthesia	27.61 (2.3)	27.10 (3.37)	0.161	27.61 (2.3)	26.5 (0.7)	0.062
Age exposure anesthesia	12.44 (14.75)	6.23 (22.59)	**0.0003**	12.44(14.75)	1.83 (2.32)	**0.003**
Number of general anaesthesia	0.22 (0.55)	3.02 (2.24)	**0.0000**	0.22 (0.55)	2.81 (1.89)	**0.000**
Total dose of sufenta received (microgram by kilogram by hour)	0.61 (0.36)	0.72 (0.46)	0.156			
Total dose of midazolam received (microgram by kilogram by hour)	29.4 (15.1)	31.7 (19.2)	0.377			
**Treatment with analgesia ****						
Number of sedations	1.44 (0.7)	0.87 (1.11)	**0.0027**	1.44 (0.7)	0.69 (1.01)	**0.0005**
Total dose of morphine received (microgram by kilogram by hour)	7.31 (3.95)	9.74 (4.92)	**0.0003**			
Duration of sedation (days)	10.3 (7.6)	9.5 (17.5)	**0.0026**	10.3 (7.6)		
**Indication for anesthesia**						
Central line placement	2 (11.1)	41 (48.2)	**0.004**			
Nasotracheal intubation	0 (0)	75 (88.2)	**0.000**			
Surgery (all indication)	2 (11.1)	36 (42.3)	**0.013**			
Patent ductus arteriosus	2 (11.1)	16 (18.8)	0.732			
Necrotizing enterocolitis	0 (0)	25 (29.4)	**0.005**			
Others indications	0 (0)	5 (5.8)	0.584			

Data expressed as *n* (%) or mean (SD); SD: standard deviation; GA: gestational age; Weeks GA: weeks of amenorrhea; * Anesthesia exposition: sevoflurane and/or another molecule (opiates and midazolam). ** Analgesia: midazolam and opiates. The PS was estimated by a multivariate logistic regression model; 2:1 propensity score matching was adopted by matching: nosocomial infection, anemia, surgery, bronchoplumanary dysplasia, patent ductus arteriosus, and retinopathy of prematurity. After matching, a logistic regression model was performed to compare the outcome measures between sevoflurane and non-sevoflurane groups. Firth’s correction was applied by performing Firth’s penalized-likelihood logistic regression to take into account the small numbers.

**Table 4 children-09-00548-t004:** Neurocognitive outcomes for children between 7 and 9 years old according to sevoflurane exposure in univariate and multivariate models (*n* = 18 without sevoflurane exposure vs. 85 exposed to sevoflurane) and with propensity score (*n* = 18 without sevoflurane exposure vs. 36 exposed to sevoflurane).

	Multivariate Model ^a^ (18 vs. 85)	Propensity Score ^b^ (18 vs. 36)
^ **1** ^ ** Cerebral palsy**	5.09 (1.18–32.99) *	3.96 (1.01–22.26) *
^ **2** ^ ** Severe disability**	3.75 (1.10–15.26) *	3.17 (0.96–12.17) *
**Visuo-spatial integration delay**	6.97 (1.21–90.67) *	5.05 (1.24–29.18) *
**Ideomotor dyspraxia**	7.50 (1.38–87.66) *	7.86 (1.60–77.92) *
**Attention deficit**	1.65 (0.46–6.90)	1.72 (0.51–6.31)
**FSIQ ≤ 85**	2.16 (0.62–8.13)	2.41 (0.70–8.61)
**WMI ≤ 85**	1.01 (0.31–3.33)	1.35 (0.41–4.59)
**PSI ≤ 85**	2.70 (0.82–9.70)	**4.20 (1.24–15.41) ***

Data expressed as adjusted OR: Odd Ratio; CI: confidence interval 95%; Weeks GAA: weeks of amenorrhea; FSIQ: Full Scale Intellectual Quotient; WMI: Working Memory index; PSI: Processing Speed Index. ^a^: pre-selected explanatory variables included in the multivariate analysis were: antenatal steroids, gestational age, gender, severe bronchodysplasia, surgery requirements, nosocomial infection, patent ductus arteriosus, anaemia, and retinopathy of prematurity. ^b^: pre-selected explanatory variables included in the propensity score were: gestational age, gender, chronic lung disease, surgery requirements, nosocomial infection, patent ductus arteriosus, anaemia, and retinopathy of prematurity; Matching 2:1 *: *p* < 0.05. ^1^ Cerebral palsy was defined as any abnormality of tone or posture according to the definition of Bax [24]. This broad definition therefore includes cerebral palsy as usually defined as well as children with only very moderate cerebral palsy (type 1 and 2 Gross Motor Function Classification System (GMCFS) and/or bimanual motor function (BMF) or ideomotor dyspaxia) who have thus benefited from neuro-psychological assessments and neuro-motor examination. ^2^ Disability was defined according to the mean of the Full-Scale Intelligence Quotient (FSIQ) and the results of the Touwen Infant neurological examination. (i) No disability: an FSIQ ≥89 and a Touwen normal; (ii) Mild disability: an FSIQ < 89 and ≥79 or a Touwen abnormal; (iii) Moderate disability: an FSIQ < 79 and ≥65 independent of the Touwen result; (iiii) Severe disability: mental delay and FSIQ < 65 independent of the Touwen result or autism, according to the DSM IV classification [27], r a severe CP not eligible for neurocognitive assessment. Severe impairment (severe cerebral palsy and severe disability).

**Table 5 children-09-00548-t005:** Neurocognitive outcome between the two groups (overall cohort and with the propensity score (PS)).

	Overall Cohort (*n*: 103)			PS-Matched Cohort (*n*: 54)		
	Non-Exposed (*n* = 18)	Exposed to Sevoflurane (*n* = 85)	*p*-Value	Non-Exposed (*n* = 18)	Exposed to Sevoflurane (*n* = 36)	*p*-Value
**WISC IV results**						
Verbal comprehension index	93.44 (21.64)	95.42 (19.43)	0.7	93.44 (21.64)	89.24(19.21)	0.34
Visual -perceptual reasoning index	87.24 (17.75)	85.95 (15.65)	0.8	87.24 (17.75)	77.07(13.13)	0.037
Working memory index	87.06 (17.14)	87.36 (14.93)	0.7	87.06 (17.14)	82.66(14.84)	0.46
Processing speed index	87.76 (16.98)	84.16 (14.46)	0.28	87.76 (16.98)	78.62(12.92)	0.05
Full scale intelligent quotient (FSIQ)	86.06 (22.51)	84.5 (17.47)	0.74	86.06 (22.51)	75.93(15.86)	0.05
**Subtest NePSY results**						
Tower	10.47 (3.26)	10.63 (2.59)	0.45	10.47 (3.26)	10.03(2.96)	0.5
Fluence of design	7.35 (3.46)	6.96 (2.46)	0.87	7.35 (3.46)	6.48(2.03)	0.49
Visual attention	9.06 (2.88)	9.02 (2.99)	0.45	9.06 (2.88)	8.54(2.88)	0.47
Auditive attention	8.93 (1.22)	9.24 (1.46)	0.82	8.93 (1.22)	9(1.54)	0.94
**Touwen results (developmental coordination disorder)**	1 (5.88)	24 (35.82)	0.015	1 (5.88)	13 (44.83)	0.05
Cerebral palsy (10 Severe CP 3 or 4 and 17 Minor CP <or =2)	3 (11)	25 (30)	0.14	3 (11)	13(33.34)	0.04
Severe disabilities	4 (23.53)	33 (39.2)	0.21	4 (23.53)	18(51.48)	0.05
Major sequelae	5 (27.78)	42 (50.00)	0.087	5(27.78)	21 (60)	0.026
**Specific cognitive impairment**						
Dysexecutive disorders	12 (70.56)	47 (70.15)	0.9	12 (70.56)	23(79.31)	0.72
Attention deficit	5 (29.41)	23 (34.85)	0.67	5 29.41	12(42.86)	0.36
Visuo-spatial integration delay	2 (12.5)	18(27.27)	0.33	2 12.5	13(46.43)	0.022
Motor ideomotor dyspraxia	1 (5.88)	21 (31.34)	0.034	1 (5.88)	12(41.38)	0.015
Language delay	4 (25)	21 (31.34)	0.76	4 (25)	14(48.28)	0.12

Data expressed as *n* (%) or mean (SD); SD: standard deviation. The PS was estimated by a multivariate logistic regression model; 2:1 propensity score matching was adopted by matching: nosocomial infection, anemia, surgery, bronchopulmonary dysplasia, patent ductus arteriosus, and retinopathy of prematurity. After matching, a logistic regression model was performed to compare the outcome measures between sevoflurane and non-sevoflurane groups. Firth’s correction was applied by performing Firth’s penalized-likelihood logistic regression to take into account the small numbers.

**Table 6 children-09-00548-t006:** Neucognitive outcome between the three groups (non-exposed, exposed to sevoflurane only, and exposed to sevoflurane and others anesthetics) in multivariate (A) and propensity score (B) model.

A. Multivariate Model ^a^			
	Non-Exposed Reference	Sevoflurane Exposure Only Odd-Ratio (CI 95%)	Sevoflurane and Others Anesthetics Odd-Ratio (CI 95%)
FSIQ ≤ 85	1	1.94 (0.51–7.83)	2.52 (0.54–13.80)
^1^ Cerebral palsy	1	3.16 (0.65–21.85)	8.88 (1.70–67.91) *
^2^ Severe disability	1	2.86 (0.77–12.37)	5.96 (1.35–32.21) *
Visuo-spatial integration delay	1	3.48 (0.52–46.27)	18.65 (2.43–293.12) *
Ideomotor dyspraxia	1	9.27 (1.44–129.41) *	6.24 (0.97–77.34)
Attention deficit	1	1.63 (0.40–7.47)	1.67 (0.36–8.74)
WMI ≤ 85	1	0.82 (0.23–2.94)	1.44 (0.35–6.42)
PSI ≤ 85	1	2.27 (0.63–8.75)	3.62 (0.83–18.43)
**B. Propensity Score Model ^b^**			
	**Non-Exposed** **Reference**	**Sevoflurane Exposure Only** **Odd-Ratio (CI 95%)**	**Sevoflurane and Others Anesthetics** **Odd-Ratio (CI 95%)**
FSIQ ≤ 85	1	1.63 (0.47–5.67)	3.30 (0.78–13.61)
^1^ Cerebral palsy	1	2.47 (0.45–13.62)	5.45 (0.96–31.13)
^2^ Severe disability	1	2.64 (0.69–10.14)	5.38 (1.31–22.06) *
Visuo-spatial integration delay	1	1.26 (0.21–7.58)	9.14 (1.49–56.04) *
Ideomotor dyspraxia	1	10.11 (1.14–90.07) *	8.26 (0.81–84.19)
Attention deficit	1	1.26 (0.34–4.66)	1.50 (0.32–6.95)
WMI ≤ 85	1	0.70 (0.19–2.54)	2.35 (0.58–9.46)
PSI ≤ 85	1	1.70 (0.49–5.93)	6.10 (1.55–23.97) *

Data expressed as adjusted OR: Odd Ratio; CI: confidence interval 95%; FSIQ: Full Scale Intellectual Quotient; WMI: Working Memory index; PSI: Processing Speed Index. *: *p* < 0.05. ^a^: pre-selected explanatory variables included in the multivariate analysis were: antenatal steroids, gestational age, gender, severe bronchodysplasia, surgery requirements, nosocomial infection, patent ductus arteriosus, anaemia, and retinopathy of prematurity. ^b^: pre-selected explanatory variables included in the propensity score were: gestational age, gender, chronic lung disease, surgery requirements, nosocomial infection, patent ductus arteriosus, anaemia, and retinopathy of prematurity; Matching 2:1 ^1^ cerebral palsy was defined as any abnormality of tone or posture according to the definition of Bax [24].This broad definition therefore includes cerebral palsy as usually defined as well as children with only very moderate cerebral palsy (type 1 and 2 Gross Motor Function Classification System (GMCFS) and/or bimanual motor function (BMF) or ideomotor dyspaxia) who have thus benefited from neuro-psychological assessments and neuro-motor examination. ^2^ Disability was defined according to the mean of the Full-Scale Intelligence Quotient (FSIQ) and the results of the Touwen Infant neurological examination. (i) No disability: an FSIQ ≥89 and a Touwen normal; (ii) Mild disability: an FSIQ < 89 and ≥79 or a Touwen abnormal; (iii) Moderate disability: an FSIQ < 79 and ≥65 independent of the Touwen result; (iiii) Severe disability: mental delay and FSIQ < 65 independent of the Touwen result or autism, according to the DSM IV classification [27], r a severe CP not eligible for neurocognitive assessment. ^3^ Severe impairment (severe cerebral palsy and severe disability).

## Data Availability

The datasets that were generated and/or analyzed during the current study are not publicly available due to the data belonging to the Assistance Publique Hopitaux de Marseille. However, datasets are available from the sponsor (promotion.interne@ap-hm.fr) on reasonable request and after signing a contract pertaining to the provision of data and/or results.

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
