# Peer review of "Long Term Neurodevelopmental Outcomes after Sevoflurane Neonatal Exposure of Extremely Preterm Children: A Cross-Sectional Observationnal Study"

_children, 2022, doi:10.3390/children9040548_

Round 1
Reviewer 1 Report
In this study, the authors ask what effects the use of Sevoflurane anesthetic during surgery performed on extremely preterm newborns during their initial hospitalization has on long-term neurodevelopmental outcomes. The authors performed a battery of motor and neuropsychological tests on a cohort of ex-preterm infants born < 28 weeks gestation between 7 and 10 years of age, and collected a large amount of clinical data from the children’s initial hospitalization. Children were grouped by Sevoflurane exposure and the primary outcome was survival with a full-scale IQ > 1 SD below the mean. The authors performed propensity score matching, employing a set of explanatory variables, and employed a 1:2 match of controls:propensity score matched exposed infantscontrols. The authors find that infants exposed to sevoflurane anesthetic have an increased risk of adverse neurodevelopmental outcomes.
The animal literature is quite convincing that exposure to almost all neuroactive medications during development have deleterious effects on neuronal survival. It is therefore surprising that studies in infants are mixed in their findings. Accordingly, a study providing clear information about sevoflurane exposure would be a welcome addition to the literature.
The validity of the conclusions drawn by the authors is lessened by two major concerns: A. which children were included in the study cohort and B. the validity of the propensity score matching. Detailed concerns follow.
- Cohort groups and cofounders
- It seems that there may be 4 groups of children in this cohort – 1) children not exposed to any anesthetics (N=36); 2) children exposed to anesthetics other than sevoflurane (N=18); 3) children exposed to anesthetics including sevoflurane; 4) children exposed to sevoflurane only. If there are children in group 4, how many were there?
- Why are children unexposed to any anesthetic not used as the controls? This is a key decision made by the authors and requires detailed justification.
- Key information is lacking on whether the two groups are comparable: What other anesthetics were the children exposed to in each group? How did the frequencies of each anesthetic differ between the sevoflurane-exposed (SEVO) and unexposed children? What are the total doses of each anesthetic each child received?
- It is unclear why children were exposed to sevoflurane or other anesthesia. Thus, in the non-exposed group, only 11% of unexposed children and 42% of SEVO children had surgery. However, as the remainder were not excluded for not having any anesthesia, they all must have had anesthesia exposure. What other indications were there for anesthetic exposure? These should be listed and the frequencies in each group shown.
- What surgical procedures requiring anesthesia of any kind were the children exposed to between the groups, and how did the frequencies of different surgical procedures differ between the groups? How do authors determine that it isn’t the severity of illness prior to surgery (NEC, PDA ligation), or the length of anesthesia (of any kind) that determines the difference in long term outcomes?
- Propensity score matching:
- Case control studies with matching typically use every case available, and match each case with one or more controls. In this study, the authors have done the opposite: all controls were used, and the cases chosen to match them based on propensity score matching. This seems backwards, and deserves clear justification.
- Much more detail is needed about each factor used to build the propensity score, as this determines the validity of the groups. A chart would be very helpful to show how each pre-selected explanatory variable was employed. For example, was gestational age continuous or dichotomized? Antenatal steroid dose – total dose, number of doses, or yes/no?
- Please show the distributions of the two groups’ propensity scores, so their comparability can be assessed.
- In table 2, it is unclear why the authors wish to present data on the overall cohort, especially for variables that did not differ significantly between the overall cohort, and which stayed non-significant in the PS-matched cohort. Table 2 could be restricted to those variables that different significantly between the original cohort and the changes in their differences after PS matching.
- On pg 11, lines 323-325, the text minimizes the differences between the unexposed group and the SEVO group: “there is a significant difference between exposed and unexposed to Sevoflurane only for modality anesthetic and analgesic treatments.” Yet, these variables may strongly affect the primary outcome, and are highly significant – other anesthesia, age at exposure (what are the values?), number of general anesthesias. These results should be explained at length in the text. Then, in the discussion, the authors state (line 437-438 ) “the multivariate and propensity score analysis results confirm that the use of another anesthetic is not a confounding factor.” This statement does not appear to be supported by the data. If the authors have additional data to support this conclusion, these data should be included.
- In Table 3, Sevoflurane treatment data is presented for the overall cohort but is missing for the PS-matched SEVO group.
- In Table 3, there are no differences in any neurosensory morbidities, hospital length, between the groups in the either overall cohort or the PS matched cohort. Inclusion of the rows in the table increases the difficulty in reading and understanding the table. Please remove these rows and include a simple statement in the text: “there were not differences between the groups in the number of abnormal cranial US or MRIs, or in length of hospitalization.”
- Table 3 is enormous, and combines variable that could contribute to the outcomes studied, as well as some of the results -WISC IV results, and following. The results portion of Table 3 should be put into its own table.
- The authors identify a primary outcome for the study, yet it is not reported. It should be stated clearly as the first finding.
- Table 4 is unclear. Why are three analyses presented? The authors have employed propensity score matching, presumably because the groups without such matching. Why therefore, are comparisons between the 18 and 85 patients presented?
Reviewer 2 Report
This paper reports on a retrospective study showing that Sevoflurane, used in EPT neonates (before 45 weeks GA corrected age), was associated with an increased occurrence of severe neurologic disorders such as cerebral palsy, severe disability or moderate impairment such as visuo-perception delay and/or ideomotor dyspraxia. Although this is a retrospective study, the results argue for the possibility of long-term effects on EPT infants who had a sevoflurane exposure before 45 weeks corrected GA.
This is a very well conducted study on an extremely important topic. Data and statistical analysis are solid and well presented. There are few suggestions
- The authors are encouraged to cite the paper by Wash et al, Surgery requiring general anesthesia in preterm infants is associated with altered brain volumes at term equivalent age and neurodevelopmental impairment, published in 2020 with similar results. (PMID: 32575110)
- Would also consider citing studies performed in nonhuman primates showing that sevoflurane and other anesthetics cause apoptosis of neurons and oligodendrocytes (PMID: 28969320, 31175984, 30951850)
- What does the phrase mean: The non-inclusion criteria were any anesthesia during the long stay: Lines 109-110?
- Please proofread the text for errors. For example, lines 201-202, outcomes should be outcome, standrat should be standard; line 342 “bronchopulmonary” is misspelled; Lines 470-472, the same sentence is repeated twice.
Author Response
Dear reviewers,
Thank you for your letter and for the reviewers’ comments on our manuscript.
All of these comments were very helpful for revising and improving our paper. We have studied these comments carefully and have made corresponding corrections that we hope will meet with your approval. The changes in the revised manuscript are marked in red. The responses to the reviewers’ comments are provided below.
We would like to express our great appreciation to you and the reviewers for the comments on our paper.
Kind regards,
Reviewer 2
- The authors are encouraged to cite the paper by Wash et al, Surgery requiring general anesthesia in preterm infants is associated with altered brain volumes at term equivalent age and neurodevelopmental impairment, published in 2020 with similar results. (PMID: 32575110)
This paper was added in the manuscript as requested.
- Would also consider citing studies performed in nonhuman primates showing that sevoflurane and other anesthetics cause apoptosis of neurons and oligodendrocytes (PMID: 28969320, 31175984, 30951850)
We’ve added these references in the manuscript.
- What does the phrase mean: The non-inclusion criteria were any anesthesia during the long stay: Lines 109-110?
We have corrected this sentence which was not meaningful.
- Please proofread the text for errors. For example, lines 201-202, outcomes should be outcome, standrat should be standard; line 342 “bronchopulmonary” is misspelled; Lines 470-472, the same sentence is repeated twice.
We have proofread the entire manuscript and corrected errors in the text.
Round 2
Reviewer 1 Report
The authors have responded to the previous review with helpful changes to the manuscript. Several easily addressed issues remain.
1. The rationale provided by the authors for excluding children unexposed to anesthetic is convincing. However, this rationale does not appear in the manuscript. It would be helpful if a summary of the rationale provided to reviewers were included either in the methods or the discussion.
2. The use of inhaled anesthetics for any procedure performed in the NICU is unheard of in North America, particularly intubation. Because of the international nature of this journal, it would be very helpful if the authors could include a sentence or table stating the indications for providing sevoflurane, and numbers and percentages of children receiving it for each indication.
3. The authors’ rationale for performing propensity score matching all of the controls and only some of the exposed babies is clearly stated in the response to the reviewers, but does not appear in the text. It should be included in the manuscript, perhaps in the methods section.
4. A 4 panel table of the distribution of propensity scores was provided for the reviewers in response to a question by a reviewer. It looks like the authors may have intended to include it as a supplementary figure (Ajouter en table supplémentaire“, but it does not appear in the manuscript. This would be very helpful.
5. The primary outcome for the study was the number of children surviving with a total FSIQ higher than 85 at school age. Because the primary outcome is the most important result, it should be presented early in the results -- after the baseline characteristics -- and presented clearly. In this study, it should be the first sentence of the first paragraph where comparisons between exposed and unexposed groups are presented: “The number of children with the primary outcome did/did not differ between the exposed and unexposed groups”, followed by the relevant counts and Odds Ratios and p value. In like fashion, the primary outcome is the first result presented and discussed in the Discussion section.
It appears that the authors did not find a significant difference in the primary outcome between the groups “The proportion of children with FSIQ ≤ 85 was similar between exposed and non-exposed group”, although this statement is the inverse of the stated primary outcome. Why do the authors find no difference in the primary outcome, yet find other differences?
